# The impact of authentic leadership on the work engagement of primary and secondary school teachers: The serial mediation role of school climate and teacher efficacy

Yanhong Shao[1], Wenxuan Jiang[2], Ningjun Wang[3], Chao Zhang[4], Lili Zhang[5]*

**1** Xiangshui Teacher Development Center, Yancheng, China, **2** School of Teacher Education, Weifang Engineering Vocational College, Weifang, China, **3** College of Education, Shandong Normal University, Jinan, China, **4** College of Foreign Languages, Qufu Normal University, Qufu, China, **5** College of Preschool Education, Qilu Normal University, Jinan, China

* zll602604@163.com

## Abstract

### Background

Teachers' work engagement significantly impacts the overall teaching quality and educational effectiveness of a school. Previous research has identified authentic leadership, school climate, and teacher efficacy as pivotal influencers that substantially shape teachers' work engagement. However, the specific mechanisms through which authentic leadership influences the work engagement of primary and secondary school teachers have not been thoroughly investigated.

### Methods

This study seeks to explore how authentic leadership shapes the work engagement of primary and secondary school teachers through the sequential mediating roles of school climate and teacher efficacy. In June 2024, an anonymous survey was successfully administered to 1043 primary and secondary school teachers ($M = 36$, $SD = 9.547$) in Shandong Province. Data analysis was performed using the Structural Equation Model (SEM) in AMOS 24.0 and SPSS 24.0.

### Results

The findings indicate that: (1) Authentic leadership directly predicts teachers' work engagement; (2) Authentic leadership influences teachers' work engagement through the mediating roles of school climate and teacher efficacy separately; (3) School climate and teacher efficacy play a chain mediating role between authentic leadership and teachers' work engagement.

**Data availability statement:** All relevant data are within the manuscript and its Supporting Information files.

**Funding:** This study was supported by Chao Zhang's International Chinese Language Education Research Program (23YH82C), the Higher Education Youth Innovation Team Project of Shandong Province (2023RW050), and the Research Project of Humanities and Social Sciences of the Ministry of Education (24YJC740084). The funder- Chao Zhang played roles in formal analysis, funding acquisition and reviewing & editing the paper.

**Competing interests:** The authors have declared that no competing interests exist.

## Conclusion

These findings contribute to a deeper understanding of the mechanisms linking authentic leadership and teachers' work engagement, and offer guidance for school administrators and policy makers to implement targeted interventions aimed at bolstering teachers' work engagement, which is essential for elevating the quality of education in schools.

## 1. Introduction

The establishment of a high-quality team of primary and secondary school teachers is a crucial aspect of basic educational reform aimed at enhancing quality and efficiency [1]. Within the process of developing such a team, teachers' work engagement has emerged as a pivotal factor for elevating the standard of education and teaching, and driving educational reform [2]. Work engagement refers to a positive and fulfilling psychological state at work, arising from the integration of three interconnected components: vigor, dedication, and absorption [3]. Vigor involves a high level of energy and psychological resilience, as well as the readiness to engage in work and persevere in the face of challenges, while dedication signifies a strong sense of commitment to work, the ability to derive a sense of purpose, passion, pride, and challenge from it [4]. Absorption denotes a state of being happily immersed in work and not easily disengaged from it [5]. Primary and secondary school teachers' work engagement not only serves to enhance the quality of teaching, stimulate students' enthusiasm for learning, and foster their growth and development, but also contributes to improving teachers' own quality of life and advancing their professional development [6]. Therefore, the study of primary and secondary school teachers' work engagement holds significant theoretical and practical implications.

The ecological systems theory posits that the school environment plays a crucial role in cultivating teachers' work engagement. In the school environment, school leadership is an important factor influencing the growth and development of teachers. Authentic leadership has been considered a suitable model of school leadership [7]. Authentic leadership is a contemporary leadership style that emphasizes genuine and values-based leadership, acknowledging and accommodating the legitimate needs of individuals, groups, organizations, communities, and cultures in an integrative manner [8]. It encompasses four dimensions: self-awareness, relational transparency, balanced processing, and concerns, and internalized moral perspective [9,10]. Authentic leadership exerts a more profound influence on teachers' work engagement compared to other forms of school leadership [11]. This enhanced impact of authentic leadership is attributed to the core tenets of authentic leadership, which prioritize personal care and support for teachers [12]. When leaders demonstrate genuine concern and encouragement, teachers are likely to experience heightened enthusiasm and motivation for their teaching roles, leading to increased work engagement. Furthermore, authentic leadership fosters a positive, supportive, and encouraging school culture environment, thereby inspiring teachers' passion and dedication to their work [13].

Research indicates that school climate is considered an important factor influencing teachers' work engagement [14]. School climate refers to the relatively enduring and stable characteristics of the school environment that can be experienced and have a significant impact on the psychological and behavioral outcomes of its members [15]. It encompasses five dimensions: instructional innovation, collaboration, decision making, student relations, and school resources [16]. Eldo and Shoshani indicated that a positive school climate typically encourages cooperation, sharing, and teamwork, enabling teachers to receive support from colleagues and leaders, thereby feeling a greater sense of belonging and satisfaction, and thus being more willing to invest in their work [17]. Studies also show that teacher efficacy affects their work engagement [18–20]. Teacher efficacy, originating from Bandura's self-efficacy theory [21], primarily refers to teacher' confidence and beliefs in their ability to successfully accomplish teaching tasks and have a positive impact. It is divided into three dimensions: efficacy for instructional strategies, efficacy for classroom management, and efficacy for student engagement [22]. When teachers believe they are capable of addressing various teaching challenges, they are more likely to maintain an optimistic attitude, actively face the difficulties in their work, and thus have more motivation to engage in their work.

While researchers have proposed that these factors influence teachers' work engagement, there is a significant lack of exploration into the specific mechanisms through which authentic leadership affects teachers' work engagement. Therefore, this study aims to integrate the Chinese context and explore the relationship between authentic leadership, school climate, teaching efficacy, and teachers' work engagement. It also seeks to analyze the mediating roles of school climate and teacher efficacy in the relationship between authentic leadership and teachers' work engagement. The study provides new empirical evidence for exploring indigenous experiences and practical pathways for cultivating teachers' work engagement with Chinese characteristics.

## 2. Literature review and research hypotheses

### 2.1 Authentic leadership and teacher work engagement

Authentic leadership has been found to positively impact teachers' work engagement [23]. Bronfenbrenner's Ecological Systems Theory posits that an individual's development is the result of interactions between the individual and their environment, which is composed of four nested levels: the microsystem, mesosystem, exosystem, and macrosystem [24]. Comprised of family, peer groups, schools, and neighbors, the microsystem constitutes the most immediate living environment for an individual [25]. Environmental factors within the microsystem can have either positive or negative impacts on an individual's development and adaptation [26]. Leaders who exhibit authenticity—characterized by behaviors such as transparency, trustworthiness, and empathy, create a positive microclimate within the school. This fosters a sense of psychological safety and support for teachers, which in turn can enhance their work engagement. Empirical studies have also indicated a positive relationship between authentic leadership and teachers' work engagement. For instance, Başaran and Kıral's survey of 300 secondary school teachers revealed that the authentic leadership exhibited by school administrators significantly predicted the teachers' reported levels of work engagement [27]. In a similar vein, Alazm and Al-Mahdy's survey of 333 teachers across 25 primary schools in Kuwait indicated that authentic leadership positively and significantly influences teachers' work engagement [2]. Echoing these findings, Kulophas et al. reported the results of a quantitative cross-sectional survey involving 605 teachers from 182 primary schools, demonstrating the impact of authentic leadership on teachers' work engagement [7]. Building on this body of evidence, the present study posits the following hypothesis:

Hypothesis 1: Authentic leadership is positively related to primary and secondary school teachers' work engagement.

### 2.2 The mediating role of school climate

A positive school climate can reduce depression and interpersonal sensitivity, increase a sense of belonging, decrease occupational burnout, stimulate work creativity, and fulfill self-worth [28]. Research indicates that authentic leadership is

more effective than traditional directive leadership in establishing a positive school climate. Authentic leadership emphasizes sincerity and transparency, allowing leaders to display their true selves and build trust and supportive relationships with teachers and staff, which contributes to an open, inclusive, and cooperative school climate. The Ecological Systems Theory underscores the interdependence between the macro-environment and micro-environments [24]. Authentic leadership may exert its influence on school climate by enhancing the relationships among teachers, students, and peers, thereby shaping the interpersonal dynamics within the educational setting. Moreover, empirical studies have shown a positive association between authentic leadership and school climate. For example, Salip and Quines suggested that authenticity in leadership plays a crucial role in creating a positive school climate [29], and Srivastava et al. indicated that authentic leadership is helpful in developing an inclusive classroom environment [30]. These findings underscore the vital role of authentic leadership in fostering a positive school climate.

School climate is closely related to teacher work engagement. Social exchange theory posits that an individual's commitment and loyalty in an organization are influenced by their exchange relationship with the organization [14]. In a positive school climate, teachers typically perceive support and recognition from school management and colleagues, which forms a social exchange relationship that motivates teachers to become more engaged in their work and improve their job performance [31]. Additionally, a positive school climate provides more resources and opportunities to meet teachers' growth and development needs, further promoting their work engagement. Empirical research also supports the close relationship between school climate and teacher work engagement. For example, a study by Song et al. involving 1,125 responses from 38 Korean high schools found that an innovative school climate significantly influences teacher work engagement [31]. Similarly, Eldor and Shoshani conducted a study with a sample of 423 teachers from 30 different schools in Israel and found that teacher work engagement was most strongly demonstrated when the service climate was high [17]. Therefore, this study proposes the following hypothesis:

Hypothesis 2: School climate mediates the relationship between authentic leadership and primary and secondary school teachers' work engagement.

## 2.3 The mediating role of teacher efficacy

Teachers with high efficacy demonstrate higher levels of self-confidence in their work and are more confident in controlling their environment and behavior, while those with low efficacy may choose to reduce effort or give up [32]. Numerous studies have demonstrated a positive correlation between teacher efficacy and their work engagement [19,20]. For instance, a longitudinal study found that teachers with a higher degree of self-efficacy are likely to show a higher level of engagement in their teaching [18]. Similarly, a study of 50 early childhood teachers found a significant positive relationship between teachers' sense of efficacy and work engagement [33]. Furthermore, a study of 614 Chinese EFL teachers observed a positive relationship between self-efficacy and work engagement [34].

Moreover, teacher efficacy is believed to be influenced by authentic leadership. Self-efficacy theory emphasizes the belief individuals hold about their capabilities to complete specific tasks, which in turn affects their behavior, motivation, and performance [30]. Authentic leadership enhances teachers' self-efficacy by granting them greater responsibility and autonomy, making them feel their work is more significant and valuable [35]. Additionally, related research has also confirmed this conclusion, showing that authentic leadership promotes teacher efficacy. Özdemir et al. conducted a survey of 756 public secondary school teachers in Turkey and found that principal authentic leadership was associated with direct effects through teacher self-efficacy [36]. Lambersky, after interviewing 20 secondary school teachers in Canada, reported a positive relationship between a principal's practices and teacher self-efficacy [37]. Adil and Kamal found in self-reported questionnaires from 500 university teachers from Pakistan that authentic leadership demonstrated a direct effect on self-efficacy [38]. Alazmi and Al-Mahdy highlighted that teachers' occupational self-efficacy plays a mediating role between authentic leadership and teachers' work engagement [2]. Although the study focused on 33 teachers in 25 primary schools

in Kuwait and limited teachers' efficacy to occupational self-efficacy, it provides valuable insights and direction for the mediation hypothesis in this research. Based on these findings, the following assumption is proposed:

H3: Teacher efficacy mediates the association between authentic leadership and primary and secondary school teachers' work engagement.

### 2.4 The chain mediating role of school climate and teacher efficacy

The school climate has a positive impact on teacher efficacy. Bandura believes that the formation and development of efficacy are influenced by verbal persuasion, vicarious experiences, emotional arousal, and mastery experiences [39]. The school work environment contains a great deal of verbal persuasion and emotional arousal content, which are important sources of teacher efficacy. A positive school collaborative atmosphere can increase mutual support among teachers, and the process of mutual support and problem-solving through peer support and open discussion contains constructive and encouraging persuasive information, which is beneficial for the construction of teachers' teaching efficacy [40]. Furthermore, in a positive school climate, teachers helping each other, collaborating, and engaging in various forms of peer evaluation and feedback can have an emotional arousal effect, increasing teachers' confidence and courage, thereby enhancing their teaching efficacy [41].

Numerous empirical studies have shown that the school climate is conducive to enhancing teacher efficacy [42,43]. Goddard and Kim found in their research that in a supportive and open school climate, teachers are more willing to learn from each other, collaborate, and improve their teaching efficacy [44]. A study of 3951 primary and secondary school teachers in Norway indicated that a positive school climate significantly promotes the enhancement of teacher efficacy [45]. Research in China has also confirmed that the school environment has a significant positive predictive effect on teacher efficacy [46]. Based on this, this study proposes research hypothesis

H4: School climate and teacher efficacy jointly plays a chain mediating role in the relationship between authentic leadership and primary and secondary school teachers' work engagement.

Based on the above literature review and research hypotheses, the theoretical model of this study is shown in Fig 1.

## 3.  Materials and methods

### 3.1  Data source and research sample

The study first determined the sampling site in accordance with the principle of convenience sampling. Following this, cluster sampling was conducted among educators from four primary schools, four secondary schools, and three high schools within a specific district of Shandong Province. Prior to the survey, the survey staff participating in the survey

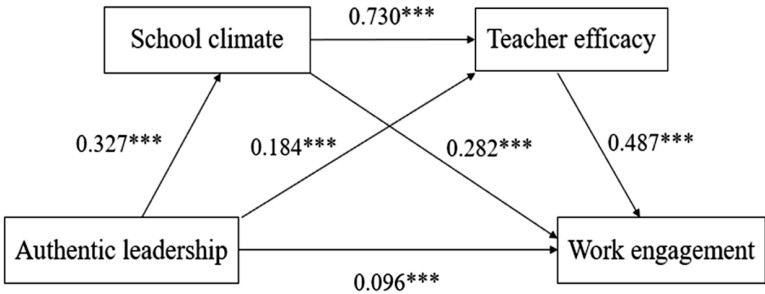

**Fig 1.  The theoretical model.**

received training to ensure their understanding of the survey's purpose and methods, mastery of survey techniques and standards, in order to enhance the quality and reliability of the survey. During the survey, teachers were briefed on the purpose and significance of the survey, and it was explained that the survey data would be kept confidential, emphasizing that participation in the survey was voluntary. We obtained written informed consent from teachers. Subsequently, the data was collected on the spot. A total of 1200 questionnaires were distributed between June 1, 2024, and June 15, 2024. After excluding questionnaires with missing values, a final total of 1,043 valid questionnaires were obtained, resulting in an effective response rate of 86.91%. Among these, high school teachers accounted for 36.7%, junior high school teachers accounted for 32.0%, and primary school teachers accounted for 31.3%. Male teachers accounted for 33.0%, and female teachers accounted for 67.0%. The age of the participants was 36 ($SD = 9.547$), with 1.7% having a college degree, 84.5% having a bachelor's degree, and 13.7% having an undergraduate degree. This study adhered to the ethical principles outlined in the Declaration of Helsinki and was granted approval by the Ethics Committee at Qufu Normal University.

### 3.2 Research instruments

**3.2.1 Authentic leadership scale.** The viewpoints of teachers regarding authentic leadership were assessed with the Authentic Leadership Scale, developed by Walumbwa et al. [47], adopted by Shapira-Lishchinsky and Tsemach [9] in educational settings. The scale comprises 14 items, categorized into The 5-point Likert scale was used, with scores ranging from 1–5 indicating "strongly disagree" to "strongly agree", with higher scores indicating higher authentic leadership. The scale has demonstrated good reliability and validity [5].

**3.2.2 School climate scale.** School climate was measured adopting the scale developed by Johnson et al. [16] and combined with the research of Liu et al. [48]. It mainly focused on cooperation (6 items, e.g., "There is good communication among teachers.") and school resources (4 items, e.g., "The school library has sufficient resources and materials.") for assessment. The questionnaire utilized a 5-point Likert scale, ranging from "1-Strongly Disagree" to "5-Strongly Agree", with higher scores indicating a more positive school climate.

**3.2.3 Teacher efficacy scale.** Teacher efficacy was assessed using Teacher Sense of Efficacy Scale (TSES), developed by Tschannen-Moran and Hoy [49].The short form contained 12 items assessing three dimensions: efficacy for instructional strategies (How well can you implement alternative strategies in your classroom?), efficacy for classroom management (How much can you do to calm a student who is disruptive or noisy?), and efficacy for student engagement (How much can you do to motivate students who show low interest in schoolwork?). Each dimension contained four items. The ratings on the Likert scale ranged from 1 (none at all) to 9 (a great deal). The reliability and validity of the scale have been validated in the previous study [50].

**3.2.4 Work engagement scale.** Teachers' work engagement was measured with the simplified version of the Work Engagement Scale, developed by Schaufeli et al. [51], was utilized to evaluate teachers' levels of work engagement. This scale consists of 9 items, encompassing three dimensions: vigor (e.g., "As soon as I wake up in the morning, I am eager to go to work."), dedication (e.g., "I am proud of the work that I do."), and absorption (e.g., "I immerse myself in my work."). Each item was assessed using a 7-point Likert scale, with scores ranging from 0 (Never) to 6 (Always), where higher scores indicate greater work engagement. The reliability and validity of the scale have been confirmed in previous research [4].

### 3.3 Statistical analysis

First, this study conducted a common method variance test on the four constructs—authentic leadership, positive school climate, teaching efficacy, and work engagement utilizing SPSS 24.0. Secondly, descriptive and correlational analyses were performed using SPSS 24.0. Thirdly, the measurement model and structural model were assessed using factor loadings, Cronbach's α, CR, AVE, and goodness-of-fit. Finally, AMOS 24.0 was employed to test the independent mediating effect and chain mediating effect of school climate on teacher efficacy.

## 4. Results

### 4.1 Common method variance

This study utilized the Harman's single-factor test to examine the presence of serious common method bias [52]. Exploratory factor analysis was conducted on all items, revealing that when unrotated, five eigenvalues were greater than 1, with the first common factor explaining 37.164% of the total variance, which is less than the 40% critical standard [53]. This indicates the absence of a serious issue of common method bias.

### 4.2 Correlation analyses

As are presented in **Table 1**, the correlational analysis indicates that authentic leadership, school climate, teacher efficacy, and work engagement are all significantly positively correlated with each other. Specifically, authentic leadership and teachers' work engagement demonstrated a significant positive relationship ($r = 0.296$, $p < 0.01$); authentic leadership and school climate exhibited a significant positive relationship ($r = 0.499$, $p < 0.01$); school climate was significantly positively related to teachers' work engagement ($r = 0.384$, $p < 0.01$); authentic leadership was significantly and positively related to teacher efficacy ($r = 0.269$, $p < 0.01$); teacher efficacy was significantly positively related to teachers' work engagement ($r = 0.645$, $p < 0.01$); and school climate significantly and positively related to teacher efficacy ($r = 0.363$, $p < 0.01$). The results of the correlational analysis among the variables indicate that further mediation tests can be conducted.

### 4.3 Assessment of measurement model and structural model

In the measurement model, the standardized factor loadings are significant and ideally above 0.50, indicating that the items are good indicators of their respective constructs [54]. The values of Cronbach's α and CR are over 0.7, indicating the acceptable reliability [55]. The AVE values surpassed the recommended threshold of 0.5, signifying satisfactory convergent validity [56]. The square root of the AVE should be greater than the correlations with other constructs, indicating that the constructs have discriminant validity [57]. As presented in Table 2, the value of Cronbach's α ranged from 0.916 to 0.965, indicating high reliability. The standardized factor loadings covered a range between 0.601 and 0.901 ($p < .001$), while the values of CR and AVE ranged from 0.918 to 0.965 and from 0.530 to 0.720 respectively, indicating acceptable convergent validity. In Table 3, the square root of AVE for each construct was greater than the correlation with other constructs, indicating acceptable levels of discriminant validity.

The structural model was evaluated using the goodness-of-fit indices. The fit indices for the structural model are as follows: $X^2/df = 1.288$ ($X^2 = 1209.63$, $df = 939$), GFI $= 0.972$, AGFI $= 0.967$, CFI $= 0.993$, TII $= 0.993$, NFI $= 0.972$, RMSEA $= 0.017$. All the values met the recommended thresholds [58], indicating a good fit for the structural model.

### 4.4 Testing for mediation effect

The study utilized the bootstrap method to investigate the mediating effects among the four constructs. Mediation is deemed statistically significant when the confidence intervals, derived from the Bias-Corrected at a 95% confidence level,

**Table 1. Descriptive statistics and correlation analysis (N = 1043).**

| | M | SD | 1 | 2 | 3 | 4 |
|---|---|---|---|---|---|---|
| 1 Authentic leadership | 3.689 | 0.919 | 1 | | | |
| 2 School climate | 3.784 | 0.602 | 0.499** | 1 | | |
| 3 Teacher efficacy | 6.292 | 1.444 | 0.269** | 0.363** | 1 | |
| 4 Work engagement | 4.786 | 1.223 | 0.296** | 0.384** | 0.645** | 1 |

Note:*$p < 0.05$,**$p < 0.01$,***$p < 0.001$, same below.

**Table 2. Evaluation of reliability and validity.**

| Latent variable | SC | P-value | Cronbach's a | CR | AVE |
|---|---|---|---|---|---|
| Authentic leadership (AL) | 0.656-0.860 | *** | 0.953 | 0.953 | 0.593 |
| School climate (SC) | 0.601-0.819 | *** | 0.916 | 0.918 | 0.530 |
| Teacher efficacy (TE) | 0.739-0.872 | *** | 0.965 | 0.965 | 0.699 |
| Work engagement (WE) | 0.729-0.901 | *** | 0.956 | 0.958 | 0.720 |

**Table 3. The test for discriminant validity of potential variables.**

| Potential variable | Authentic leadership | School climate | Teacher efficacy | Work engagement |
|---|---|---|---|---|
| Authentic leadership | **0.770** | | | |
| School climate | 0.535*** | **0.728** | | |
| Teacher efficacy | 0.276*** | 0.376*** | **0.836** | |
| Work engagement | 0.307*** | 0.412*** | 0.672*** | **0.848** |

Note: The square root of the AVE of four latent constructs is given in the diagonal, and the correlation coefficient is given on the below diagonal.

exclude zero [59]. Data analysis was conducted employing Amos 24.0 software. The findings pertaining to the mediating roles of school climate and teacher efficacy between authentic leadership and work engagement are delineated in Table 4. The direct effect of authentic leadership on teachers' work engagement is significant (**Unstandardized coefficients** = 0.096, $P < 0.05$), supporting the acceptance of hypothesis one. Positive school climate and teacher efficacy mediate the relationship between authentic leadership on teachers' work engagement, with a total indirect effect of 0.298. Specifically, the indirect effect is composed of three pathways: The pathway of authentic leadership →school climate →teacher efficacy→ work engagement had an indirect effect of 0.116 with a 95% confidence interval of [0.081, 0.161]; The pathway of authentic leadership →school climate→ work engagement had an indirect effect of 0.092 with a 95% confidence interval of [0.028, 0.162]; The pathway of authentic leadership→teacher efficacy→work engagement had an indirect effect of 0.089 with a 95% confidence interval of [0.034, 0.162]. The Bootstrap method yielded 95% confidence intervals for all three indirect effects that do not encompass zero, signifying that each of the indirect effects is statistically significant. This finding substantiates the hypotheses 2, 3, and 4.

Additionally, this study assessed the indirect effect percentages of school climate and teacher efficacy as partial mediators. As presented in Table 4, of the three significant indirect effects, the school climate's mediation represents 31% of the total indirect effect, and teacher efficacy's mediation represents 30%. Moreover, the combined indirect effect of school climate and teacher efficacy constitutes 39% of the total indirect effect, representing the most substantial mediating role. The specific mechanisms by which authentic leadership influences teachers' work engagement through the pathways of school climate and teacher efficacy are illustrated in Fig 2.

### 4.5 Robustness test

**4.5.1 Sobel verification.** Based on the Sobel test results (see Table 5), the indirect effects of the paths AI→SC→WE and AI→TE→WE are both significantly present. Specifically, the indirect effect of the path AI→SC→WE is 0.092 (SE = 0.024, z = 3.823, $p < 0.001$), accounting for 23.4% of the total effect (0.092/ 0.394); the indirect effect of the path AI→TE→WE is 0.206 (SE = 0.026, z = 7.923, $p < 0.001$), accounting for 52.3% of the total effect (0.206/ 0.394). The combined mediating effects of the two paths account for 75.7% of the total effect (0.092 + 0.206 = 0.298/ 0.394), and when combined with the direct effect (0.096, accounting for 24.3% of the total effect), the results indicate that the mediating

**Table 4. Mediating effects of school climate and teacher efficacy (N = 1043).**

| Path relationship | | Effect | SE | 95% confidence internal | |
|---|---|---|---|---|---|
| | | | | Lower | upper |
| **Test of indirect, direct and total effects** | | | | | |
| DistallE | AL→SC→TE→WE | 0.116 | 0.021 | 0.081 | 0.161 |
| SCIE | AL→SC→WE | 0.092 | 0.033 | 0.028 | 0.162 |
| TEIE | AL→TE→WE | 0.089 | 0.033 | 0.034 | 0.162 |
| TIE | Total indirect effect | 0.298 | 0.043 | 0.206 | 0.375 |
| DE | AL→WE | 0.096 | 0.053 | 0.003 | 0.215 |
| TE | Total effect | 0.394 | 0.051 | 0.296 | 0.501 |
| **Percentage of indirect effects** | | | | | |
| P1 | DistallE/TIE | 0.390 | 0.087 | 0.260 | 0.572 |
| P2 | SCIE/TIE | 0.310 | 0.091 | 0.108 | 0.478 |
| P3 | TEIE/TIE | 0.300 | 0.093 | 0.132 | 0.482 |
| P4 | TIE/TE | 0.756 | 0.119 | 0.513 | 0.990 |
| P5 | DE/TE | 0.244 | 0.119 | 0.010 | 0.487 |

Note: AL=Authentic leadership, SC=School climate, TE=Teacher efficacy, WE=Work engagement, Standardized estimating of 1000 bootstrap sample, ***p<0.001.

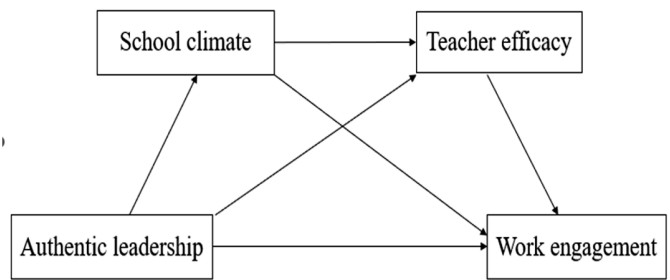

**Fig 2. The path diagram.**

effects dominate the total effect. This is consistent with the conclusions of the Sobel-Goodman test [60], further validating the robustness of the research model. It suggests that authentic leadership (AL) has a significant indirect impact on work engagement (WE) through school climate (SC) and teacher efficacy (TE), with each path having a distinct mechanism of action. Among them, the effect of Al→TE→WE is more pronounced, occupying the majority of the mediating effect proportion.

**4.5.2 Retest.** To ensure the scientific validity of the research findings, this study conducted a robustness test by substituting the original independent variable. Authentic Leadership (AL) is intricately linked with School Climate (SC), Teacher Efficacy (TE), and Work Engagement (WE), and is considered to have enduring significance for teachers' professional performance. Consequently, the study opted to replace the initial independent variable with the sub-dimension variable of Authentic Leadership, specifically internalized moral perspective (IMP), and carried out regression analysis to assess the model's robustness. As depicted in Table 6, when compared to the baseline regression outcomes, the model's significance and the directionality of the regression coefficients are consistent, with only nominal variations in the precise values of the coefficients. This consistency suggests that the research conclusions exhibit a substantial level of robustness.

**Table 5. Sobel−Goodman mediation test result.**

|  | Path | Effect | SE | z-value | p-value |
|---|---|---|---|---|---|
| Sobel1 |  | 0.090 | 0.028 | 5.826 | 0.001 |
| Indirect effect | AI→SC→WE | 0.092 | 0.024 | 3.823 | 0.001 |
| Direct effect | AI→WE | 0.096 | 0.036 | 2.681 | 0.007 |
| Total effect | AI→WE | 0.394 | 0.040 | 9.958 | 0.001 |
| Sobel2 |  | 0.206 | 0.026 | 7.923 | 0.001 |
| Indirect effect, | AI→TE→WE | 0.206 | 0.026 | 7.771 | 0.001 |
| Direct effect | AI→WE | 0.096 | 0.036 | 2.681 | 0.007 |
| Total effect | AI→WE | 0.394 | 0.040 | 9.958 | 0.001 |
| Sobel3 |  | 0.116 | 0.016 | 7.251 | 0.001 |
| Indirect effect, | AI→SC→TE→WE | 0.116 | 0.016 | 7.295 | 0.001 |
| Direct effect | AI→WE | 0.298 | 0.031 | 9.613 | 0.001 |
| Total effect | AI→WE | 0.414 | 0.033 | 12.545 | 0.001 |

**Table 6. The result is expressed as a function of the sub-dimensions of X.**

|  | WE |  |  | SC |  |  | TE |  |  | WE |  |  |
|---|---|---|---|---|---|---|---|---|---|---|---|---|
|  | SE | β | 95%CI | SE | β | 95%CI | SE | β | 95%CI | SE | β | 95%CI |
| Constant | 0.12 | – |  | 0.056 | – |  | 0.142 | – |  | 0.192 | – |  |
| IMP | 0.031 | 0.258** | 0.200~0.0410 | 0.014 | 0.385** | 0.166~0.222 | 0.037 | 0.236** | 0.213~0.357 | 0.026 | 0.064** | 0.015~0.116 |
| SC |  |  |  |  |  |  |  |  |  | 0.054 | 0.15** | 0.024~0.091 |
| TE |  |  |  |  |  |  |  |  |  | 0.021 | 0.575** | 0.446~0.529 |
| $R^2$ | 0.066 |  |  | 0.148 |  |  | 0.056 |  |  | 0.445 |  |  |
| Adjusted $R^2$ | 0.065 |  |  | 0.147 |  |  | 0.055 |  |  | 0.444 |  |  |
| Adjusted F | 73.305** |  |  | 179.353** |  |  | 60.742** |  |  | 275.653** |  |  |

Note: * and ** indicate significance at the statistical levels of 10% and 5%, respectively. IMP = Internalized moral perspective.

## 5. Discussion

Empirical evidence suggests that authentic leadership, school climate, and teacher efficacy has a positive impact on teachers' work engagement. However, there is still a gap in understanding the specific mechanisms through which authentic leadership impacts teachers' work engagement via school climate, and teacher efficacy. This research aimed to construct a mediation model to investigate whether authentic leadership would be indirectly correlated with teachers' work engagement through school climate, and teacher efficacy. The findings are presented as follows.

The results of the study revealed that authentic leadership was directly and positively related to teachers' work engagement, which not only corroborates Cortés-Denia et al.'s research [23], which establishes a positive correlation between authentic leadership and teachers' work engagement, but also resonates with Srivastava and Dhar's proposition [11] that teachers' work engagement is significantly influenced by leadership styles, with authentic leadership exerting a particularly profound impact. From the perspective of the Ecological Systems Theory, we can gain a deeper understanding of how individual behavior is influenced by the social environment. As Li has observed, authentic leadership is crucial in shaping individuals' intrinsic motivation and their ensuing professional dedication [61]. The recognition by teachers of their leaders' authenticity, transparency, and investment in their professional advancement is instrumental in fostering a sense of intrinsic motivation, which subsequently amplifies their work engagement [62]. Furthermore, teachers who are recipients

of robust support and encouragement from their leadership are less likely to be distracted and more inclined to be actively immersed in their professional endeavors [63]. This study reinforces the proposition that authentic leadership serves as a significant predictor of work engagement among teachers.

The results of the study identified school climate as one significant partial mediating role between authentic leadership and teachers' work engagement. This finding is consistent with previous research that has suggested a positive relationship between authentic leadership and school climate [28,29], as well as a positive association between school climate and teachers' work engagement [17,30]. These findings support the notion that authentic leadership plays a crucial role in shaping the perception of school climate, which, in turn, affects teachers' engagement in work. Besides, the finding aligns with the principle of the Ecological Systems Theory that the interaction between individuals and their environment is a dynamic developmental process [24]. Authentic leadership can continuously interact and engage in feedback loops, adjusting and optimizing leadership strategies to adapt to changes in school climate and teachers' needs, thereby promoting teacher work engagement and the improvement of the school atmosphere. The finding provides more evidence of the role of school climate in the relationship between authentic leadership and teachers' work engagement.

The results of the study demonstrated that teacher efficacy plays a significant partial mediating role between authentic leadership and teachers' work engagement. This finding is in line with Alazmi and Al-Mahdy's perspective, which emphasizes the crucial role of teacher efficacy in the association between authentic leadership and teachers' work engagement [2]. On one hand, authentic leadership encourages teachers to participate in decision-making and provides opportunities for successful experiences, thereby boosting their self-efficacy. On the other hand, authentic leadership empowers teachers with more autonomy, making them feel that their work is more significant and valuable, which in turn enhances their self-efficacy. When teachers possess a strong sense of self-efficacy, they are more likely to engage proactively in their work, demonstrating resilience in the face of challenges, and thereby increasing their work engagement. This study reiterates the importance of teacher efficacy in the relationship between authentic leadership and teachers' work engagement.

The results of the study further showed that both school climate and teacher efficacy functioned as a chain mediating role between authentic leadership and teachers' work engagement.

Authentic leadership emphasizes authentic expression, giving teachers more confidence and encouraging mutual trust and cooperation among them. This leadership style is conducive to building a positive school climate of trust, cooperation, and equitable dialogue. In this atmosphere, teachers support each other, share ideas and practices, thereby enhancing their teacher efficacy and ultimately promoting teachers' work engagement. This finding further elucidates the mechanisms by which environmental systems and individual factors influence teachers' work engagement and advances the previous research by shedding light on how authentic leadership can increase teachers' work engagement. It is worth noting that although both school climate and teacher efficacy as mediators were established, their effect sizes were 0.31 and 0.30, respectively, which were lower than the effect of serial mediation. This indicates that the serial mediation of school climate and teacher efficacy has a more significant impact on teachers' work engagement. This suggests that when intervening in teachers' work engagement at the school level, cultivating school climate and teacher efficacy should be given greater priority. The implications of these findings resonate beyond local boundaries, offering insightful lessons for educational practices across China and serving as a template for educational reforms on a global scale. By integrating the principles of authentic leadership, educational institutions worldwide can create environments that foster teacher efficacy and commitment, thereby enhancing the quality of education for all students.

## 6. Limitations and future research directions

This study has some limitations. Firstly, the use of cross-sectional data, while enabling the identification of influences between variables, makes it difficult to establish clear causal relationships between the variables. Future research could employ longitudinal data to delve deeper into these relationships. Secondly, constrained by data limitations, this study only analyzed the mediating pathways of school climate and teacher efficacy in the impact of authentic leadership on teachers'

work engagement. Subsequent research should refer to relevant theories to conduct a more detailed exploration of other potential pathways through which authentic leadership influences teachers' work engagement. Lastly, as this study was primarily based on samples from Shandong, the external validity of the conclusions is somewhat limited. Future research could validate the findings through surveys in a more diverse range of regions to ensure that the conclusions are more compelling and applicable in real-world settings.

## 7. Implications

The findings of this study hold significant implications for theoretical understanding and practical application in addressing the work engagement of primary and secondary school teachers.

From a theoretical perspective, this study contributes to the existing literature by elucidating the chain mediating role of positive school climate and teacher efficacy in the relationship between authentic leadership and work engagement. It advances our understanding of the complex mechanisms underlying this association and emphasizes the importance of considering multiple factors in explaining work engagement among primary and secondary school teachers. This study underscores the significance of school climate and teacher efficacy as pivotal mediators in this relationship, shedding light on the underlying processes that can be targeted for intervention and prevention efforts. It offers guidance for future research to explore the mechanisms of these mediating variables across diverse educational and cultural settings, and helps expand the theoretical framework of authentic leadership's impact on work engagement. This prompts researchers to consider additional mediating or moderating variables for a more comprehensive understanding of the relationship's dynamics.

From a practical standpoint, this study offers valuable insights for designing effective interventions to enhance work engagement among primary and secondary school teachers. Interventions could focus not only on promoting authentic leadership but also on cultivating positive school climate and teacher efficacy. To improve authentic leadership, on one hand, leaders could deeply analyze their personal values, beliefs, and moral standards to ensure that they consistently adhere to these core principles in their leadership, motivating faculty and staff to pursue meaningful work lives [64]. On the other hand, leaders could maintain open, honest, and transparent communication with teachers. They should listen to their voices, respect their opinions, and fully consider their needs and expectations in the decision-making process. Recognizing the pivotal role of a positive school climate in enhancing teacher work engagement, educational institutions must prioritize its development within the constraints of limited resources. Strategic allocation of funds is crucial for optimizing both the physical teaching environment, which directly impacts comfort and workspace quality, and the intangible aspects that contribute to teacher well-being and job satisfaction. Balancing capital expenditures on infrastructure with operational costs, particularly teacher salaries, is essential to prevent adverse budgetary trade-offs. An integrated approach that sustains competitive remuneration while upgrading the physical campus is imperative. Additionally, deliberately cultivating campus culture through initiatives such as cultural walls and display boards, which highlight institutional and individual achievements, can significantly bolster teachers' sense of belonging and pride. To foster teacher efficacy, the principal could passionately promote authentic leadership, consistently reinforce a culture of open communication and collaboration, and spark teachers' internal motivation for professional growth by fostering shared leadership experiences and providing personalized professional development. Ultimately, this will boost teachers' dedication and teaching performance. Additionally, schools could offer targeted assistance to address practical issues encountered in teaching through means such as teacher professional development training and school-based professional learning communities, in order to alleviate teaching pressures and enhance teachers' sense of efficacy. These practical insights provide specific ideas and directions for future research on how to more effectively design and implement intervention measures to improve teacher work engagement, prompting researchers to pay attention to the combined effects of different intervention measures and their applicability in different educational stages and teacher groups.

 

## Supporting information

**S1 File. S1 dataset.**
(XLSX)

## Acknowledgements

The authors wish to thank Guanghai Cao for providing technical support in data analysis for this research.

## Author contributions

**Conceptualization:** Yanhong Shao, Lili Zhang.

**Data curation:** Wenxuan Jiang, Ningjun Wang.

**Formal analysis:** Chao Zhang.

**Funding acquisition:** Chao Zhang.

**Investigation:** Yanhong Shao, Wenxuan Jiang, Ningjun Wang, Lili Zhang.

**Methodology:** Wenxuan Jiang, Lili Zhang.

**Writing – original draft:** Yanhong Shao.

**Writing – review & editing:** Wenxuan Jiang, Ningjun Wang, Chao Zhang, Lili Zhang.

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
