## [Decision Letter · Decision Letter 0]

19 Nov 2024

PONE-D-24-26731The impact of authentic leadership on the work engagement of primary and secondary school teachers: The serial mediation role of school climate and teacher efficacyPLOS ONE

Dear Dr. Shao,

Thank you for submitting your manuscript to PLOS ONE. After careful consideration, we feel that it has merit but does not fully meet PLOS ONE’s publication criteria as it currently stands. Therefore, we invite you to submit a revised version of the manuscript that addresses the points raised during the review process.

We look forward to receiving your revised manuscript.

Kind regards,

Md Nazirul Islam Sarker, PhD

Academic Editor

PLOS ONE

Journal requirements:    When submitting your revision, we need you to address these additional requirements. 1. Please ensure that your manuscript meets PLOS ONE's style requirements, including those for file naming. The PLOS ONE style templates can be found at https://journals.plos.org/plosone/s/file?id=wjVg/PLOSOne_formatting_sample_main_body.pdf and https://journals.plos.org/plosone/s/file?id=ba62/PLOSOne_formatting_sample_title_authors_affiliations.pdf 2. PLOS requires an ORCID iD for the corresponding author in Editorial Manager on papers submitted after December 6th, 2016. Please ensure that you have an ORCID iD and that it is validated in Editorial Manager. To do this, go to ‘Update my Information’ (in the upper left-hand corner of the main menu), and click on the Fetch/Validate link next to the ORCID field. This will take you to the ORCID site and allow you to create a new iD or authenticate a pre-existing iD in Editorial Manager. 3. Thank you for stating the following financial disclosure:  [This work was supported by the International Chinese Language Education Research Program under Grant 23YH82C. This work was supported in part by the Higher Education Youth Innovation Team Project of Shandong Province under Grant 2023RW050].  Please state what role the funders took in the study.  If the funders had no role, please state: ""The funders had no role in study design, data collection and analysis, decision to publish, or preparation of the manuscript."" If this statement is not correct you must amend it as needed. Please include this amended Role of Funder statement in your cover letter; we will change the online submission form on your behalf. 4. We note that the grant information you provided in the ‘Funding Information’ and ‘Financial Disclosure’ sections do not match.  When you resubmit, please ensure that you provide the correct grant numbers for the awards you received for your study in the ‘Funding Information’ section. 5. Please include your full ethics statement in the ‘Methods’ section of your manuscript file. In your statement, please include the full name of the IRB or ethics committee who approved or waived your study, as well as whether or not you obtained informed written or verbal consent. If consent was waived for your study, please include this information in your statement as well.  6. Please include captions for your Supporting Information files at the end of your manuscript, and update any in-text citations to match accordingly. Please see our Supporting Information guidelines for more information: http://journals.plos.org/plosone/s/supporting-information. 

Additional Editor Comments:

The author is advised to address all comments point-by-point.

Reviewers' comments:

Reviewer's Responses to Questions

**Comments to the Author**

1. Is the manuscript technically sound, and do the data support the conclusions?

Reviewer #1: Yes

Reviewer #2: Partly

Reviewer #3: Yes

2. Has the statistical analysis been performed appropriately and rigorously? 

Reviewer #1: Yes

Reviewer #2: Yes

Reviewer #3: Yes

3. Have the authors made all data underlying the findings in their manuscript fully available?

Reviewer #1: Yes

Reviewer #2: Yes

Reviewer #3: Yes

4. Is the manuscript presented in an intelligible fashion and written in standard English?

Reviewer #1: Yes

Reviewer #2: Yes

Reviewer #3: Yes

5. Review Comments to the Author

Reviewer #1: Comments to the Author PONE-D-24-26731

1.The research significance of this paper needs to be analyzed in depth, especially in the abstract, where it should be clearly stated rather than using vague phrases like “practical implications”, “crucial role”. For example, in what specific ways can the work engagement of primary and secondary school teachers be improved?

2.The discussion needs to be analyzed from a more critical perspective. I noticed that much of Section 5 is simply a description of the previous hypotheses, without presenting many novel or critical viewpoints. For example, the author mentions that this study is primarily based on a sample from Shandong Province, China, which raises concerns about the external validity of the conclusions. This is an important point. The author needs to analyze the significance of this study for China as a whole, and even globally, by comparing the basic characteristics of different provinces in China. If the study’s findings are largely applicable only to Shandong Province, it would be more appropriate to publish it in a scholarly journal in Shandong rather than in Plos One.

Another typical example is the statement in the discussion: “schools should invest resources to improve the physical environment of the campus...”. This is a very general summary. I believe that schools have limited resources, and these resources may be in competition with one another. For instance, if some funds are allocated to improving the physical environment, then teachers’ salaries may be correspondingly reduced. How could this situation be evaluated? Of course, I agree that improving the physical environment is important, but the premise is that the project should have clear goals and a balance with other indicators. The author needs to explain this aspect. In other words, the author not only needs to state the benefits of improving the physical environment but also needs to acknowledge the potential risks involved.

At the same time, I noticed that the author used many uncertain terms to describe the research findings, such as “One possible reason is that authentic leadership ignites teachers’ intrinsic motivation, leading them to develop a deeper love for their profession.” However, this is not a characteristic of a high-quality paper. The author needs to use clear language to present the research findings, rather than using vague statements. The author should clearly explain how authentic leadership influences teachers’ motivation, which requires a solid literature review. Additionally, I noticed that the author frequently uses “should” to describe conclusions, but if we consider the practice scenario, ‘could’ would be more appropriate. If ‘should’ is used, there needs to be sufficient arguments to prove that it is reasonable.

3.Authentic leadership is the focus of this paper's discussion, but after reading it, I believe it leans more towards “political discourse”, as the author’s definition is quite vague. The author not only needs to discuss the dimensions used to measure authentic leadership but also needs to clarify the reasons for the emergence of authentic leadership. In other words, the author should explain the relationship between authentic leadership and the social system, how the social system fosters authentic leadership, and how authentic leadership drives the social system.

Additionally, the author states, “The social-ecological systems theory posits that individual behaviors are influenced by multiple interconnected levels of their environment, ranging from the immediate setting (microsystem) to the broader societal context (macrosystem)”. I must seriously point out that this is not the main viewpoint of the social-ecological systems theory; this concept is more aligned with sociology. I recommend the author study the papers by McCay (2002) and Giddens (1984), which indicate that while individuals are influenced by the structures in which they are embedded, they can also influence and change these structures.

I also need to correct an obvious mistake made by the author: the social-ecological systems theory is an academic term in natural resource management, proposed by Elinor Ostrom, primarily discussing the coevolution of social systems and natural systems, with a focus on polycentric governance. Clearly, the use of “the social-ecological systems” in this paper is an inappropriate concept. I suggest that the author reorganize the content of the literature review and maintain a rigorous attitude towards specialized terminology.

I have noticed that the author's analysis primarily states that principals need to possess authentic leadership. However, this skill is something that every leader, including grassroots leaders, should have.

4.In the empirical section, I suggest the author include a “Robustness test” section to verify whether the research results remain stable and reliable under different conditions. This actually addresses my second comment, where the author needs to ensure that the findings are valid under various conditions and can be applied to regions beyond Shandong. The robustness test should primarily include “Sobel verification” and retesting. For example, the author can refer to the method used by Zhao et al. (2023) in discussing the mediation effect.

Reference

[1] McCay B. J. (2002). “Emergence of institutions for the commons: contexts, situations, and events,” in The Drama Of The Commons. Eds. E. Ostrom E., Dietz T., Dolšak N., Stern P. C., Sonich S., Weber E. U. (National Academy Press, Washington DC, USA), 361–402.

[2] Giddens A. (1984). The constitution of society: Outline of the theory of structuration (Cambridge, MA: University of California Press).

[3] Zhao, Y., Song, Z., Chen, J., & Dai, W. (2023). The mediating effect of urbanisation on digital technology policy and economic development: Evidence from China. Journal of Innovation & Knowledge, 8(1), 100318.

Reviewer #2: Dear authors

Congratulations on the research, I truly believe the study is of very high impact. though i also advise you reassess and write the study to gain its full impact.

You have high compiled evident to research but have not imputed it in effective manner, the implication of you study will bring forth changes in viewing teachers and their training to the current educational system. Wishing you the best.

regards

Reviewer #3: Very good study having wider implications in policy making. Authentic leadership is the key to ensure quality education in schools. Intrinsic motivation motivated through authentic leadership definitely matters in teacher's work engagement. Appreciate the interest taken in performing this study.

6. PLOS authors have the option to publish the peer review history of their article (what does this mean? ). If published, this will include your full peer review and any attached files.

**Do you want your identity to be public for this peer review?** For information about this choice, including consent withdrawal, please see our Privacy Policy .

Reviewer #1: No

Reviewer #2: No

Reviewer #3: No

---

## [Author Response · Author response to Decision Letter 0]

28 Nov 2024

Dear editor and reviewers. We extend our deepest gratitude for the insightful and valuable feedback you have provided. Your expertise and suggestions have been instrumental in enhancing the quality of our manuscript. We have carefully reviewed each of your comments and have made the necessary corrections in accordance with your guidance. We have loaded our point-by-point responses to the comments from each esteemed reviewer.

---

## [Decision Letter · Decision Letter 1]

14 Jan 2025

PONE-D-24-26731R1The impact of authentic leadership on the work engagement of primary and secondary school teachers: The serial mediation role of school climate and teacher efficacyPLOS ONE

Dear Dr. Shao,

Thank you for submitting your manuscript to PLOS ONE. After careful consideration, we feel that it has merit but does not fully meet PLOS ONE’s publication criteria as it currently stands. Therefore, we invite you to submit a revised version of the manuscript that addresses the points raised during the review process.

We look forward to receiving your revised manuscript.

Kind regards,

Md Nazirul Islam Sarker, PhD

Academic Editor

PLOS ONE

Journal Requirements:

Additional Editor Comments:

The author is advised to address the comments of the reviewer point-by-point.

Reviewers' comments:

Reviewer's Responses to Questions

**Comments to the Author**

1. If the authors have adequately addressed your comments raised in a previous round of review and you feel that this manuscript is now acceptable for publication, you may indicate that here to bypass the “Comments to the Author” section, enter your conflict of interest statement in the “Confidential to Editor” section, and submit your "Accept" recommendation.

Reviewer #1: All comments have been addressed

Reviewer #2: All comments have been addressed

Reviewer #3: All comments have been addressed

2. Is the manuscript technically sound, and do the data support the conclusions?

Reviewer #1: Yes

Reviewer #2: Yes

Reviewer #3: Yes

3. Has the statistical analysis been performed appropriately and rigorously? 

Reviewer #1: Yes

Reviewer #2: N/A

Reviewer #3: Yes

4. Have the authors made all data underlying the findings in their manuscript fully available?

Reviewer #1: Yes

Reviewer #2: Yes

Reviewer #3: Yes

5. Is the manuscript presented in an intelligible fashion and written in standard English?

Reviewer #1: Yes

Reviewer #2: Yes

Reviewer #3: Yes

6. Review Comments to the Author

Reviewer #1: The overall manuscript already meets the publication requirements of PLOS ONE, but I believe there is still an area that needs improvement. The authors need to provide definitions for key terms in the introduction section (or at the point of their first mention). For example, the term "Authentic Leadership" appears in the introduction of the first chapter, but its definition is only provided in the literature review section of Chapter 2. This is clearly inconsistent with academic conventions. The authors should define this term in the introduction, while the literature review should focus on research related to this term.

The authors may refer to their approach to defining "Work Engagement" in the introduction section as an example.

It is recommended that the authors review the entire manuscript to identify similar issues. Any newly introduced terms should be defined at their first appearance.

Reviewer #2: Dear Authors

Th revision has be inclusive of the comments and suggestions made, this presents a better clarity and hopefully provides better implications for the study in the future

Best wishes

Reviewer #3: (No Response)

7. PLOS authors have the option to publish the peer review history of their article (what does this mean? ). If published, this will include your full peer review and any attached files.

**Do you want your identity to be public for this peer review?** For information about this choice, including consent withdrawal, please see our Privacy Policy .

Reviewer #1: No

Reviewer #2: No

Reviewer #3: No

---

## [Author Response · Author response to Decision Letter 1]

16 Jan 2025

We extend our deepest gratitude for the insightful and valuable feedback you have provided. Your expertise and suggestions have been instrumental in enhancing the quality of our manuscript. We have carefully reviewed each of your comments and have made the necessary corrections in accordance with your guidance. Below, we present our point-by-point responses to the comments from each esteemed reviewer:

Reviewer 1

Reviewer’s comment: The authors need to provide definitions for key terms in the introduction section (or at the point of their first mention).

Our answer: Thank you for your positive feedback and valuable insights.We have provided definitions for key terms in the introduction section. Please take a look at the red text on Pages 2-3 in the introduction section.

Reviewer 2

Reviewer’s comment: Hopefully provides better implications for the study in the future.

Our answer: Thank you for your positive feedback and valuable insights. We have added the content on Pages 15-16 in the section of implication.

Reviewer 3

Reviewer’s comment: No Response

Our answer: We truly appreciate your positive assessment and the time you’ve taken to review our work.

---

## [Decision Letter · Decision Letter 2]

26 Feb 2025

The impact of authentic leadership on the work engagement of primary and secondary school teachers: The serial mediation role of school climate and teacher efficacy

PONE-D-24-26731R2

Dear Dr. Shao,

We’re pleased to inform you that your manuscript has been judged scientifically suitable for publication and will be formally accepted for publication once it meets all outstanding technical requirements.

Kind regards,

Md Nazirul Islam Sarker, PhD

Academic Editor

PLOS ONE

Additional Editor Comments (optional):

The author is advised to keep in touch with the production team for the remaining process of publication.

Reviewers' comments:

Reviewer's Responses to Questions

**Comments to the Author**

1. If the authors have adequately addressed your comments raised in a previous round of review and you feel that this manuscript is now acceptable for publication, you may indicate that here to bypass the “Comments to the Author” section, enter your conflict of interest statement in the “Confidential to Editor” section, and submit your "Accept" recommendation.

Reviewer #1: All comments have been addressed

Reviewer #2: All comments have been addressed

2. Is the manuscript technically sound, and do the data support the conclusions?

Reviewer #1: Yes

Reviewer #2: Yes

3. Has the statistical analysis been performed appropriately and rigorously? 

Reviewer #1: Yes

Reviewer #2: Yes

4. Have the authors made all data underlying the findings in their manuscript fully available?

Reviewer #1: Yes

Reviewer #2: Yes

5. Is the manuscript presented in an intelligible fashion and written in standard English?

Reviewer #1: Yes

Reviewer #2: Yes

6. Review Comments to the Author

Reviewer #1: The paper has met the criteria for publication in Plos One. Good luck to the authors in developing more analysis based on this, it's an interesting direction.

Reviewer #2: Dear Authors,

The paper is well balanced from my perspective. though the limitations on any research is a given, hope the paper gives trail to future research All the best!

7. PLOS authors have the option to publish the peer review history of their article (what does this mean? ). If published, this will include your full peer review and any attached files.

**Do you want your identity to be public for this peer review?** For information about this choice, including consent withdrawal, please see our Privacy Policy .

Reviewer #1: **Yes: ** Jian Chen

Reviewer #2: No

---

## [Editor Report · Acceptance letter]

PONE-D-24-26731R2

PLOS ONE

Dear Dr. Shao,

I'm pleased to inform you that your manuscript has been deemed suitable for publication in PLOS ONE. Congratulations! Your manuscript is now being handed over to our production team.

Kind regards,

on behalf of

Dr. Md Nazirul Islam Sarker

Academic Editor

PLOS ONE